# 3D registration of remote sensing data for planetary exploration

Loïs Brun, Adeline Paiement
Université de Toulon, Aix Marseille Univ, CNRS, LIS, Marseille, France
`firstname.lastname@lis-lab.fr`

Sylvain Douté
IPAG, CNRS, Université Grenoble-Alpes, Grenoble, France
`firstname.lastname@univ-grenoble-alpes.fr`

Jeronimo Bernard-Salas
ACRI-ST, Grasse, France
`firstname.lastname@acri-st.fr`

## Abstract

*Data registration is a crucial preliminary step in exploiting complementary remote sensing data from different satellites. In planetary science, registration is notoriously difficult due to a lack of salient landmark features and strong discrepancies in the spatial extent (denoted as footprint) of acquisitions. We propose to address this issue using 3D information derived from pairs of images, available as Digital Terrain Models (DTMs). We propose a method based on 3D geometric information for the rigid registration of DTMs. Our method is comparatively evaluated on a new benchmark dataset of synthetic planetary DTMs, as well as on real Martian data, against baseline and state-of-the-art methods. It demonstrates a satisfactory accuracy, while proving significantly more robust to common remote sensing footprint challenges such as missing data and varying amounts of overlap.*

## 1. Introduction

Planetary exploration and Earth remote sensing generate vast overlapping satellite datasets, offering opportunities for cross-referencing information across modalities and acquisition dates, for richer terrain analysis and detection of subtle surface features. However, sensor and measurement imprecisions lead to decameter to hectometer misalignments. Automatically registering the images is an unsolved challenge, particularly for scenes with few salient points for matching like Martian dune fields. We propose to leverage the 3D information embedded in the imagery.

A large literature on photogrammetry and photoclinometry e.g. [2, 3, 5, 19, 20] have focused on creating altitude maps, known as Digital Terrain Models (DTMs) and analogous to point clouds, from pairs of 2D observations. They typically use observations from a same instrument captured on a same orbit, when the target terrain is viewed from two angles in rapid succession. This limits the amount of misalignment between the images, hence permitting a decent quality construction of the DTM[1].

The aforementioned misalignments undermine the coherence of multi-source and multi-orbit DTMs, imposing heavy manual registration processing to planetary scientists. Automatic 3D registration methods are needed to scale to large datasets. They must address the specific challenges of DTMs. These include a smaller spatial extension in the vertical dimension compared to the horizontal dimensions, different resolutions and levels of details, and different spatial extents, denoted as *footprints*, from varying levels of overlap and possibly missing data. In addition, mostly flat terrains contain only few salient features.

Point cloud registration methods are mostly designed for 3D objects or small (often indoor) scenes with similar extents in all 3 dimensions e.g. [12, 22]. Simultaneous Localization And Mapping (SLAM) methods e.g. [6, 24] were applied to larger outdoor scenes with open boundaries. This scenario is closer to our 3D remote sensing case, although the density of salient features may differ. These methods include pairwise point-cloud registration (ICP and its many variants e.g. [1, 4, 14]), which is sometimes used in planetary science, but with limited accuracy, feature-based methods e.g. [7, 21], and deep-learning approaches e.g. [10, 18]. Planetary science applications require reproducible, interpretable, and physically grounded solutions, while lacking ground-truth registration data. For these reasons, in this study, we do not consider learning-based approaches. While terrestrial or synthetic data could be used to pre-train learning methods, they are too far from planetary science data to remove the need for fine-tuning on planetary ground-truth data, as was demonstrated in [9]. Artificial data may not exhibit the complex and structured geological features of

---

[1] Although the misalignments are not as much a problem in DTM creation than in registration of multi-source and multi-orbit DTMs, it may also benefit from an improved image registration in the future.

general planetary data, in particular in the Martian case.

Myronenko and Song proposed Coherent Point Drift (CPD) [13], a probabilistic framework for rigid and non-rigid point cloud registration that addresses the low density of salient features. CPD formulates alignment as a Gaussian mixture model (GMM) fitting problem. CPD handles large point-clouds and is a state-of-the-art (SOTA) method in the presence of noise, outliers, and missing data, which particularly fits the challenges of DTM registration.

To further address outlier-robust registration, Yang et al. [23] introduced TEASER++, a fast algorithm that reformulates the problem using a Truncated Least Squares cost that is robust to spurious matches. It solves scale, rotation, translation estimation in cascade and certifies global optimality, yielding robustness to over 99% outliers. This property may also be of interest for DTMs.

To our knowledge, neither of these two SOTA were tested on DTMs and their peculiar geometric and statistical properties. In addition, they are limited to two point clouds at a time, while planetary data commonly require combining 3+ acquisitions with different overlaps (footprints) into larger dense mosaics (process known as *mosaicking*).

In [17], Paiement et al. proposed IReSISD (Integrated Registration, Segmentation, and Interpolation of Sparse Data), a joint 3D registration and modelling method for tomographic data, that addresses the lack of matching features due to a very limited overlap. Registration is informed by the 3D information provided by a geometric model that is built from the combined data. A reformulation was proposed in [15] to any number of 3D point clouds of compact objects. As will be discussed in Sec. 2.2, this method is not readily usable on open-boundary objects such as DTMs.

We propose IReSISD-DTM, an extension of IReSISD specifically tailored to DTMs. It is particularly suited to cases with limited overlap (either due to footprint or to very different imaging resolutions), as it relies on a geometric product rather than direct data-to-data correspondences. It may handle any number of DTMs. Both registration and production of the geometric model do not rely on any extra features but solely on the DTMs.

The contributions of this work are: 1) 3D-based registration for remote sensing data, instead of the traditional 2D registration, 2) IReSISD-DTM (Sec. 2), a SOTA registration method for DTMs, tested on synthetic and real data (Sec. 4), and robust to different footprints and resolutions, 3) a synthetic planetary DTM benchmark dataset (Sec. 3).

## 2. Method

With IReSISD-DTM[2], we propose to address the registration challenges raised by planetary remote sensing data by

---

[2]The code can be found at: https://gitlab.lis-lab.fr/lois.brun/iresisd-dtm_3d4science.git

exploiting the 3D information that they contain. DTMs may sample a same surface in very different ways, with possibly a low overlap between them. We solve jointly the geometric (or topographic) modelling and the registration tasks, which share information and support each other.

We are not concerned with the creation of the DTMs from the input sensor data, which constitutes a preliminary step. Methods for DTM creation, commonly used by planetary scientists, include [2, 3, 5, 19, 20].

### 2.1. Pre-requisite: IReSISD

IReSISD-DTM is inspired by IReSISD [17], an iterative method based on the level-set approach, where a contour $C$ segments the object to be modeled from its background, under the joint influence of all data being registered. The level-set method may adapt to data of various types and modalities (images or point clouds) given an appropriate definition of the contour's evolution speed $S$. In [16, 17], an implicit interpolation of $S$ and $C$ was introduced, that allows dealing with sparse data, making the method suitable to data with highly heterogeneous resolutions. At a given registration position of the data, the converged contour $C$ of the level-set method would represent a consensus on object shape from all data at their current position. Simultaneously to the iterative evolution of $C$, the data are progressively rigidly aligned to $C$. Therefore, $C$ serves as the shared reference for aligning all (any number of) data.

Formally, contour $C$ is defined as the zero-level of a smooth scalar function $\phi$, typically a signed distance function, which is defined in the data domain (2D grid for images, 3D grid for tomographic images and point clouds). At each iteration, IReSISD computes scalar speed fields $S^n$ associated to each data $n$, and combined into a global speed $S = \sum_n S^n \circ g^n$ that drives the deformation of $\phi$ and $C$ as:

$$\frac{d\phi}{dt} = [(\delta_\epsilon(\phi) \cdot S) \star \psi] \star \psi + \text{sign}(\phi)(1 - \|\nabla\phi\|) \quad (1)$$

where $g^n(\mathbf{x}) = R^n \mathbf{x} + \mathbf{T}^n$ is the registration transform of data $n$, $\delta_\epsilon$ is an approximate Dirac function of width $\epsilon$, which effectively selects the zero-level contour $C$, $\psi$ is a radial basis function (RBF), and $\star$ denotes convolution, implemented with a Fast Fourier Transform (FFT). The recommended RBF is an inverse multiquadric $\psi(\mathbf{x}) = (\|\mathbf{x}\|^2 + \gamma^2)^{-\frac{\beta}{2}}$ with default $\beta$ being the number of dimensions in the data, so 3 in our case. $\gamma$ is to be set according to the amount of interpolation needed. The second term in Eq. (1) enforces that $\phi$ remains a signed distance function.

The definition of $S^n$ depends on the data type (tomographic images, point clouds, etc.) and on the segmentation task. Negative/positive $S$ values locally induce a retraction/expansion of $C$ along its normal, respectively. $S^n$ may not be defined everywhere in the domain, and interpolation of $S$ is achieved by the convolutions with $\psi$.

$S^n$ also serves to compute the rigid registration to be applied to data $n$ by considering the (inverse) average displacement that $n$ seeks to induce on $C$:

$$\frac{\partial T_i^n}{\partial t} = \frac{1}{|\Omega_{\text{reg}}^n|} \int_{\Omega_{\text{reg}}^n} (S^n \circ g^n)\langle \mathbf{u_i}, R^n \mathbf{N}\rangle \tag{2a}$$

$$\frac{\partial \theta_i^n}{\partial t} = \frac{1}{|\Omega_{\text{reg}}^n|} \int_{\Omega_{\text{reg}}^n} (S^n \circ g^n)\langle \tilde{R}_i^n \mathbf{x} + \mathbf{T}^n, R^n \mathbf{N}\rangle \tag{2b}$$

with $\tilde{R}_i^n$ a modification of $R^n$ with $\frac{\pi}{2}$ added to the $i^{th}$ angle $\theta_i^n$, $T_i^n$ the $i^{th}$ component of $\mathbf{T}^n$, $\mathbf{u_i}$ the unit vector in direction $i$, $\mathbf{N}$ the normal of $\phi$, and $|.|$ denoting cardinality. $\Omega_{\text{reg}}^n$ is the registration domain of $n$, which may be either the intersection $\Omega_n \cap C$ between data domain $n$ and $C$ (local variant, more accurate), or its extension $(\Omega_n \cap C) \cup \Omega_{\text{diff}}^n$ to also include $\Omega_{\text{diff}}^n$ where $S^n$ and $\phi$ are of opposite signs (global variant, more robust to local minima).

The method iterates until a stopping criterion based on the stability of $C$ and $g^n$, or for a fixed number of steps.

## 2.2. Modelling planetary surfaces

A major difference between the modelling tasks of IReSISD and IReSISD-DTM is the geometric nature of the object to segment. IReSISD was designed for compact objects whose 3D contours are closed and contained in the considered volume. In contrary, IReSISD-DTM is concerned with planetary surfaces with only one terrain-sky interface. In the longitude and latitude directions, terrains have open boundaries. This may cause issues to IReSISD which assumes a pseudo-periodicity of the considered volume due to the use of an FFT in Eq. (1). A buffer around the volume may deal with minor mismatches between opposite longitude or latitude boundaries. We found empirically that a buffer of 20 voxels for an RBF flatness parameter $\gamma$ at 1 is generally sufficient. However, the "sky" and "ground" volume boundaries strongly disagree, as they are respectively outside and inside the modelled terrain object, which correspond to opposite signs of $\phi$ at high absolute values. This drastic mismatch may hinder the convergence of $C$ due to violating the pseudo-periodicity assumption of the FFT in Eq. (1).

To overcome this limitation, we introduce an artificial bottom interface to the modeled terrain, forming some sort of "floating island" that thickens the DTM along the altitude axis. Since only the terrain's surface is relevant for studying topography, the modelling results are insensitive to the exact definition of this additional interface. The only requirement is that the terrain's thickness is sufficient everywhere to allow an unhindered propagation of $C$. We define this bottom interface at a fixed depth from the terrain's surface. We determined empirically that a depth of three time the maximum variation of the DTM's altitudes (i.e. $3(z_{\text{max}} - z_{\text{min}})$) ensures a sufficient thickness everywhere while maintaining the object at a reasonable size for computations.

Additionally, for exceptionally flat terrains, we allow a user-defined vertical-exaggeration factor to boost the effective thickness and further enhance robustness.

## 2.3. DTM-specific definition of the level-set speeds

As seen in Sec. 2.1, the level-set speeds $S^n$ are the sole drivers both for data registration and for geometric model building through the evolution of $C$. We define them based on the DTMs geometric information. No other source of information contributes to the registration or model building.

For DTMs, as for point clouds in [15], we compute $S^n$ along "sensor-to-point" sightlines. Unlike in [15], the sensor location (i.e. satellite orbit) is sufficiently far above the terrain for sightlines to be considered parallel. In addition, the notions of segmenting an object from a background, and of visible and hidden faces of the object, do not apply.

$S^n$ is defined as $-1$ outside the terrain, $1$ inside the terrain, and with a linear transition on a width $\upsilon$ around the terrain's interfaces. Formally, at a position $\tau$ along the sightline, with $\tau_U$ and $\tau_B$ denoting the locations of the upper and lower (from Sec. 2.2) interfaces:

$$S^n(\tau) = \begin{cases} -1 & \tau \leq \tau_U - \upsilon \\ \frac{\tau - \tau_U + \upsilon}{\upsilon} - 1 & \tau_U - \upsilon \leq \tau < \tau_U + \upsilon \\ 1 & \tau_U + \upsilon \leq \tau < \tau_B - \upsilon \\ 1 - \frac{\tau - \tau_B + \upsilon}{\upsilon} & \tau_B - \upsilon \leq \tau < \tau_B + \upsilon \\ -1 & \tau \geq \tau_B + \upsilon \end{cases} \tag{3}$$

$\upsilon$ assures a smooth transition to reduce the risk of $C$ oscillating around the object interfaces. A general recommendation is to set $\upsilon$ between 4 and 6 voxels. The computations for different sigthlines are independent from each other and may be parallelized on GPU in a ray tracing manner.

During registration, especially for DTMs of different resolutions, there is no guaranty that all sighlines cross voxels at their center. For higher accuracy, when computing $S$, the different $S^n$ are interpolated linearly to the neighboring voxels. When multiple sightlines from a same point cloud contribute to a same voxel (possible depending on the chosen volume resolution), their $S^n$ values are averaged.

In case of heterogeneous resolutions and gaps, voxels not intersected by any sightline are not assigned any speed $S^n$. The interpolation of $C$ in Eq. (1) deals with any gaps in $S$.

## 2.4. DTM registration

The registration scheme of IReSISD, defined in Eq. (2), may generally be applicable to DTMs, although with one major adjustment on the registration domain. When DTMs sample portions of a terrain that do not fully overlap, registration information may only exist in the overlapping areas. Indeed, in places where a single DTM contributes to the geometry modelling, $C$ adapts fully to this DTM. It is only when $C$ reflects a compromise between concurrent

DTM information that $C$ may drive registration efficiently. Thus, for the computation of the registration updates $\frac{\partial T^n}{\partial t}$ and $\frac{\partial \theta^n}{\partial t}$, we only consider the speeds $S^n$ in the subdomain $\Omega_{n \cap m}$ where DTM $n$ overlaps with at least one other DTM $m$. Formally, $\Omega_{\text{reg}}^n = \Omega_{n \cap m} \cap C$ (local variant) or $\Omega_{\text{reg}}^n = \Omega_{n \cap m} \cap (C \cup \Omega_{\text{diff}}^n)$ (global variant) in Eq. (2).

As stated above, DTMs are much more extended in the longitude-latitude dimensions than in the altitude dimension. This causes a scale difference between the $x, y$ and $z$ components respectively in Eq. (2a) which may hinder convergence. This situation may be addressed by adjusting the time step separately for $x, y$ and for $z$. In practice, we achieve this by applying the IReSISD piecewise-linear normalization (see [17]) separately to $\frac{\partial}{\partial t}\begin{pmatrix} T_x^n \\ T_y^n \end{pmatrix}$ and to $\frac{\partial T_z^n}{\partial t}$. Additionally, we improve convergence by registering along the $x, y$ directions alone for 5 iterations before a full 6-degrees-of-freedom (DoF) registration.

During registration, and similar to [17], we find beneficial to subtract an offset to each $\frac{\partial T_i^n}{\partial t}$ and $\frac{\partial \theta_i^n}{\partial t}$ update, to limit the transformations to the minimum necessary. Unlike in [17], we define these offsets as the average updates across DTMs. These offsets do not change the relative displacements between DTMs, hence they do not modify their registration as such. However, they limit the general amount of displacement with regard to $C$, which, in turn, requires minimal adjustments to the new DTM positions.

### 2.5. Initialization and speed of convergence

IReSISD-DTM being an iterative method through gradient-descent optimisation, its computation efficiency depends on the number of iterations until convergence. It is therefore interesting to initialize $C$ as close as possible to its final shape and location. We take advantage of the fact that the data's initial misalignment is limited compared to the spatial extent of the terrain. Therefore, the average shape of the unregistered point clouds yields a reasonable initial consensus geometric model. This imperfect model is then refined iteratively by IReSISD-DTM, requiring far fewer iterations than the default IReSISD's spherical initialization would.

To obtain the consensus model, we first construct signed distance functions to the interfaces of each thick (see Sec. 2.2) DTM. The average of these distance functions yields a zero-level contour $C_0$ that represents the "mean shape" of the DTMs, and that serves to compute the initial $\phi_0$ as its signed distance function.

For very sparse point clouds, the distance to the nearest data point may be high over large regions of the volume. To avoid this introducing spurious contours, we compute the distance function only along sight-lines connecting the "sensor" to the data points, then we interpolate the values using a convolution with the RBF as is done for $S$ in Eq. (1).

### 2.6. Scalability for large terrains

The user has control over the size and resolution of the volume where $\phi$ and $C$ are defined. When working with DTMs of heterogeneous resolutions, adopting that of the finest DTM allows working at the best possible resolution.

When dealing with particularly large terrains, a multi-scale approach helps to improve both computational and convergence speeds. The initial resolution of the volume may be chosen low, and increased after convergence at the next level, until the required resolution is reached. Note that the DTMs do not need to be modified or downsampled by the user, since their positioning in the common volume is dealt with when computing $S$ from $S^n$ (see Sec. 2.3).

## 3. Data

We illustrate the application of IReSISD-DTM with the case study of Martian DTMs. These datasets are representative of the properties and artifacts that may be encountered in the context of planetary remote sensing. The spatial resolution can vary widely, from submeter to hundred of meters, additionally resulting in uneven sampling and often partial overlap. Martian data also share challenges with Earth remote sensing, while being affected by higher uncertainties, thus allowing to assess the limit performance of compared methods.

Groundtruth registration is unavailable in the Martian case due to no absolute reference. We perform quantitative comparisons on synthetic datasets. Real Martian DTMs are used to validate the applicability and behavior of compared methods on real data. The same principles and conclusions would readily extend to other planetary bodies.

### 3.1. Synthetic data

We create and release publicly[3] a dataset of artificial DTMs that match, in its different subsets, the properties of real planetary science DTMs, i.e. noise level, data gaps, heterogeneous resolution and sampling, and partial overlap.

To reproduce the fractal properties of natural topography without vegetation or buildings, we use the method of [11] that synthesizes DTMs by enforcing a prescribed power-law spectrum in the Fourier domain. Control parameters are root-mean-square roughness $\sigma_{\text{ele}}$ that controls the variance of elevation, Hurst exponent $H$ which controls how "rough" or "smooth" the fractal surface will look, physical extent $L_x$, grid resolution $m \times n$, and optional roll-off wave vector $q_r$ which controls the maximum $x$ and $y$ scale of surface features. The power spectral density of the resulting DTM is computed and compared with the reference spectrum for validation. Moreover, a welcomed option is

---

[3]The project code can be found at: https://gitlab.lis-lab.fr/lois.brun/iresisd-dtm_3d4science.git, as well as the created dataset: https://zenodo.org/records/20117117

applying a spectral anisotropy, a directional bias, thereby better mimicking the linear ridges and troughs characteristic of real geology. This method enables the generation of synthetic terrains spanning diverse surface morphologies.

We vary different parameters in order to assess their effect on the registration. Each time, other parameters are kept fixed at plausible values: a terrain span of 10 000 m in length, with a $256 \times 256$ pixel grid (so $\sim$ 39m per voxel), $\sigma_{\text{ele}}$ at 10 voxels and $H$ at 0.5 to model a surface that is neither too smooth nor too rough, and $q_r$ set to zero. Since the synthetic DTMs are generated stochastically, each realization creates a different DTM including when having the same set of generating parameters.

To better emulate the imperfections found in real planetary science data, we also introduce zero-mean Gaussian noise in the height component, data gaps and partial overlap and heterogeneous spatial resolutions and associated levels of details. These perturbations are introduced individually in their related experiments, with the exception of the noise that is used at a moderate level ($\sigma_{\text{noise}} = 1$ voxel) in two experiments (Sections 4.2 and 4.3). Perturbations are added independently to each DTM being registered.

### 3.2. Real data

We experiment with two example terrains from real observations, as illustrated in Figs. 4 and 3. MOLA (Mars Orbiter Laser Altimeter, MGS) provides near-global topography at $\sim$300 m/pixel. CTX (Context Camera, MRO) delivers high-resolution grayscale images at $\sim$6 m/pixel over swaths $\sim$30 km wide. Its stereo pairs allow DTMs at $\sim$18 m/voxel. CaSSIS (Colour and Stereo Surface Imaging System, TGO) captures narrower swaths $\sim$ 10 km wide with colour and stereo imagery at $\sim$4.5 m/pixel, producing DTMs at $\sim$12 m/pixel. Together, these datasets combine global coverage with fine-scale surface detail.

## 4. Experimentations

We compare our registration against baselines for 3D point clouds, namely ICP [1] and RANSAC [6][4], and the SOTA methods CPD [13][5] and TEASER++ [23][6]. Although our method is inspired by [15], we may not compare against it since [15] is designed for compact objects and is not readily applicable to open-boundary DTMs. As discussed in Section 1, we do not compare against learning methods due to a lack of real ground-truth data. We use the standard and advised hyperparameters for every algorithm, and the global variant of IReSISD-DTM with $\gamma = 1$ and 30 iterations.

---

[4]For both baseline methods, we use the implementation from SciPy.

[5]We use the implementation from https://probreg.readthedocs.io/en/latest/.

[6]We use the implementation from https://github.com/MIT-SPARK/TEASER-plusplus.

Experiments on synthetic DTMs register two identical DTMs with optional perturbations (see Section 3.1) and a random translation of norm 20 voxels applied to one copy. This large misalignment for remote sensing data aims at establishing the limits of performance of compared methods. Although no rotation misalignment is introduced, registration is performed freely on translation and rotation. Voxel size is defined according to the finest DTM resolution. Consequently, the registration results are tied to the resolution of the input DTMs. Detailed specifications of the experiments are in the sup. materials.

Quantitative results are presented as translation and rotation errors, the latter measured by computing the rotation axis (unreported) and amount of residual rotation $\theta$. Metrics are averaged over 30 realizations of stochastic terrains and random misalignments to report mean and std values.

### 4.1. Effect of elevation variance on registration

One of the main challenges in registration lies in the availability and quality of informative features. In DTM registration, such features are predominantly related to elevation. It is therefore of interest to assess the behavior of the methods when varying the level of (un)flatness of the terrains. Although real acquisitions combine multiple confounding factors simultaneously, isolating elevation variability provides a principled lower bound on registration difficulty that allows establishing the limit performance as a function of this parameter. This is achieved by varying the elevation standard deviation $\sigma_{ele}$ of the terrain generation method in even values from 2 to 30 voxels. Results are reported in Tab. 1 and in Tab. 5 of the supplementary materials for a more complete table.

ICP and RANSAC do not perform well, with translation errors of the order of 16 voxels, likely due to the lack of clear 3D salient features and too few correspondences. Rotation error stays to 0 thanks to a successful plane to plane registration. TEASER++ fails completely with errors consistently above 200 voxels, thus we only present its results in the sup. materials. This failure may also occur because DTMs tend to lack distinct salient features. CPD achieves close to perfect results, with voxel-level to sub-voxel accuracy, even for low elevation variance and associated very flat terrains. Our method achieves comparable results to CPD, although generally only at voxel-level and sub-degree accuracy. It struggles more with the flattest case ($\sigma_{\text{ele}} = 2$, rarely achieved in real data), with an error of 3.2 voxels and 0.9°.

Throughout this experiment and the next, the rotation error std remains relatively high for our method, indicating some moderate instability for rotation alignment. This might be due to the fact that convergence is obtained much more rapidly for translation than for rotation. This warrants further investigation in the future, possibly including further delaying rotation with regard to translation as was done in

Table 1. Effect of elevation variance on registration. Translation/rotation errors are reported as mean,std over 30 independent runs, in voxels and degrees. Best results are highlighted in bold.

| $\sigma_{ele}$ | ICP | RANSAC | CPD | Proposed |
|---|---|---|---|---|
| 2 | 16.7,2.0/0.0,0.0 | 16.4,1.8/0.0,0.0 | **1.5,2.2/0.0,0.0** | 3.2,2.6/0.7,0.6 |
| 4 | 16.3,1.7/0.0,0.0 | 16.0,1.9/0.0,0.0 | 1.5,3.5/0.0,0.0 | **1.5,2.6/0.4,0.6** |
| 6 | 16.4,2.0/0.0,0.0 | 16.4,2.0/0.0,0.0 | **0.5,0.2/0.0,0.0** | 1.1,0.4/0.2,0.7 |
| 8 | 16.3,1.8/0.0,0.0 | 16.4,1.7/0.0,0.0 | 1.6,4.7/0.0,0.0 | **1.0,0.5/0.1,0.7** |
| 10 | 16.3,1.3/0.0,0.0 | 16.3,2.1/0.0,0.0 | **0.5,0.2/0.0,0.0** | 1.0,0.3/0.2,0.7 |
| 20 | 16.7,2.2/0.0,0.0 | 16.9,1.9/0.0,0.0 | **0.6,0.2/0.0,0.0** | 0.9,0.4/0.5,0.7 |
| 30 | 16.2,2.0/0.0,0.0 | 16.6,1.9/0.0,0.0 | 1.5,3.2/0.0,0.0 | **1.0,0.4/0.1,0.6** |

Table 2. Effect of missing data on registration as a function of number # and radius $r$ (in voxels) of gaps, and total gap surface $s$ (in voxel$^2$). Presentation is as in Tab. 1.

| Configuration | CPD | Proposed |
|---|---|---|
| #2 r5 s157 | 1.9, 3.2 / 0.0, 0.0 | **1.4, 1.5 / 0.2, 0.6** |
| #5 r10 s1571 | **2.5, 4.4 / 0.0, 0.0** | 2.7, 4.3 / 0.1, 0.7 |
| #20 r5 s1571 | 2.3, 3.3 / 0.0, 0.0 | **2.0, 2.7 / 0.1, 0.6** |
| #2 r30 s5655 | 5.1, 2.1 / 0.0, 0.0 | **3.6, 6.9 / 0.1, 0.6** |
| #40 r5 s3142 | 1.7, 0.7 / 0.0, 0.0 | **1.5, 1.1 / 0.1, 0.7** |
| #10 r20 s12566 | 5.7, 3.3 / 0.0, 0.0 | **2.4, 3.4 / 0.1, 0.6** |
| #1 r80 s20106 | 54.3, 35.4 / 0.0, 0.1 | **3.9, 5.4 / 0.1, 0.6** |
| #1 r120 s45239 | 162.7, 101.2 / 0.2, 0.4 | **4.4, 6.1 / 0.2, 0.6** |

[17]. Still, the demonstrated voxel-level accuracy is already satisfactory for most planetary exploration applications.

### 4.2. Effect of missing data on registration

Gaps in the data are a common challenge in remote sensing, especially with cloud coverage as it is frequent in Earth observation. To test their effect, we introduce random (circular) gaps to both registered DTMs, at independent locations. We vary both the number # and radius $r$ of holes, as reported in Tab. 2. Large holes (80 or 120 voxels) are extreme cases, unlikely in real planetary data, to serve as a stress test for benchmarking. The small-hole cases (5-30 voxels) are more realistic and demonstrate the expected behavior of the methods in challenging real acquisition conditions.

ICP, RANSAC, and TEASER++ underperform as in the previous experiment, so we omit them from Tab. 2. CPD's performance degrades markedly. While it maintains a reasonable accuracy of ∼2 voxels in cases with smaller gaps (5-10 voxels), it exhibits some notable failure modes with an erroneous rotation of the registered plane, resulting in an increased error std. CPD is particularly sensitive to the size of gaps, with dramatic failure when confronted with a single large hole of 80 voxels (Fig. 1) or 120 voxels (mean error of 162.7±101.2 voxels). Even in the more realistic scenarios, errors increase to ∼5 voxels for holes of 20 to 30 voxels.

On the other hand, IReSISD-DTM maintains a stable accuracy for all individual sizes and total surface fractions of gaps, over both realistic and more extreme benchmark conditions. It only sees a moderate increase in errors, from ∼1.5 voxels for holes of 5 voxels to 4.4 ± 6.1 voxels for the 120 voxels holes. This superior robustness to missing data is a key advantage for real applications.

### 4.3. Effect of varying levels of overlap

Overlap at different locations is a key challenge in real data. In planetary remote sensing, images are commonly acquired as large bands parallel to the track of the observing satellite, with a varying amount of overlap between consecutive tracks depending on orbit property. Therefore, registration methods need to be robust to the varying, and sometime lim-

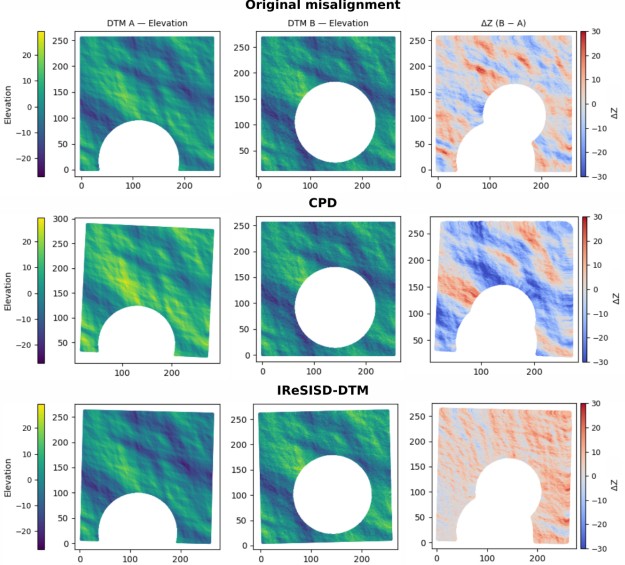

Figure 1. Robustness to important missing data showcase – single 80-voxel holes at independent locations. Left to right: Registered DTMs (color denotes elevation in voxel), (projected) elevation difference in voxels. Top to bottom translation/rotation errors in voxels/degrees: 14.76/0.00, 11.24/2.77, 1.79/2.29

ited, amount of overlap between the data acquisitions. We assess the impact of overlap at five different levels, as listed in Tab. 3 and Tab. 7 of sup. materials. Overlap is varied by truncating the DTMs in the $x$ dimension (see Fig. 2).

As in the previous experiment, ICP and RANSAC do not perform well, with no rotation error but translation errors of the order of 16 voxels, likely due to the lack of clear 3D salient features and plane to plane registration. TEASER++ fails completely with errors consistently above 200 voxels, thus we present its results only in the sup. materials. The accuracy of CPD degrades significantly as the overlap decreases, down to translation errors of 106.3 ± 4.3 voxels for 20% overlap. As seen in Fig. 2, this failure is due to CPD aligning footprints, disregarding the DTMs' contents and

Table 3. Effect of different overlaps. Presentation is as in Tab. 1.

| Overlap | ICP | RANSAC | CPD | IReSISD-DTM |
|---|---|---|---|---|
| 100% | 16.6,1.8 / 0.0,0.0 | 16.5,2.0 / 0.0,0.0 | 2.6,5.8 / 0.0,0.0 | **1.0,0.4 / 0.6,0.7** |
|  | 6.1,0.1 / 6.1,0.1 | 6.1,0.1 / 6.1,0.1 | 1.3,1.1 / 1.5,1.1 | **1.4,0.4 / 1.8,0.4** |
| 80% | 16.0,1.8 / 0.0,0.0 | 16.5,2.1 / 0.0,0.0 | 15.7,3.1 / 0.0,0.0 | **1.0,0.4 / 0.4,0.7** |
|  | 6.1,0.2 / 6.2,0.1 | 6.1,0.2 / 6.1,0.2 | 4.6,1.0 / 4.7,1.0 | **1.5,0.4 / 1.8,0.4** |
| 60% | 16.6,1.9 / 0.0,0.0 | 16.5,1.7 / 0.0,0.0 | 47.0,5.5 / 0.0,0.0 | **1.0,0.5 / 0.3,0.7** |
|  | 6.0,0.2 / 6.1,0.2 | 6.1,0.2 / 6.1,0.2 | 7.8,0.9 / 7.9,0.9 | **1.5,0.5 / 1.8,0.5** |
| 40% | 16.2,1.9 / 0.0,0.0 | 16.8,2.0 / 0.0,0.0 | 78.9,5.9 / 0.0,0.0 | **1.6,3.2 / 0.3,0.7** |
|  | 6.1,0.3 / 6.1,0.3 | 6.1,0.3 / 6.1,0.3 | 10.1,1.9 / 10.1,1.9 | **1.4,0.4 / 1.8,0.4** |
| 20% | 17.1,2.0 / 0.0,0.0 | 16.3,2.0 / 0.0,0.0 | 106.3,4.3 / 0.0,0.0 | **1.6,2.6 / 0.4,0.7** |
|  | 6.1,0.5 / 6.2,0.5 | 6.1,0.5 / 6.1,0.5 | nan,0.0 / nan,0.0 | **1.6,0.4 / 2.0,0.5** |

Table 4. Effect of heterogeneous resolution, presented as in Tab. 1.

| Downsampling factor | CPD | Proposed |
|---|---|---|
| 2 | **0.5, 0.2 / 0.0, 0.0** | 0.9, 0.2 / 0.2, 0.7 |
| 5 | **0.6, 0.3 / 0.0, 0.0** | 1.6, 0.2 / 0.6, 0.7 |
| 10 | **1.5, 0.4 / 0.0, 0.0** | 7.3, 5.7 / 0.8, 0.7 |

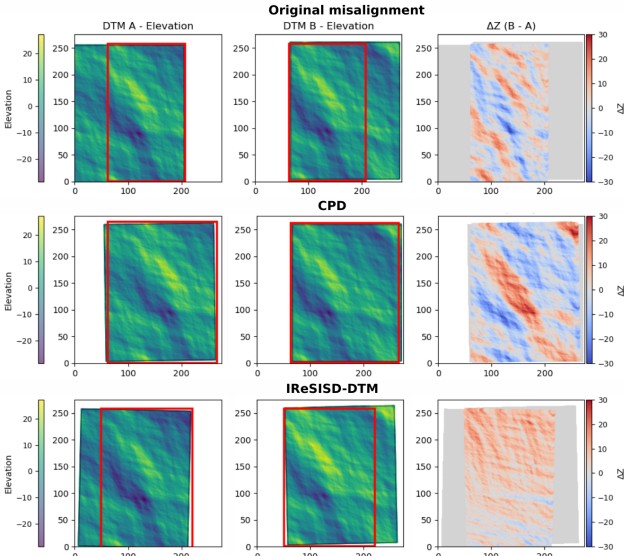

Figure 2. Illustration of a partial overlap scenario. Opposed 20% regions of each DTM are pruned, resulting in a total overlap of 60%. Left to right as in Fig. 1, with the overlapping region highlighted by a bold red rectangle. Top to bottom translation/rotation errors in voxels/degrees: 15.81/0.00, 44.53/0.89, 0.81/2.34

informative elevation differences. This is consistent with the excellent results CPD obtained in previous experiments where the DTMs' footprints were identical (Section 4.1) or very close (Section 4.2 with small holes), and degraded results in Section 4.2 with large holes. The fact that CPD fails when footprints differ significantly is a major drawback for real planetary science applications.

In contrast, our method consistently achieves voxel-level precision even in low-overlap cases. It restricts the alignment to the overlapping regions, and compensates the low availability of matching features with the exploitation of the full geometric information in these regions. This robustness is particularly valuable for planetary datasets, where similar footprints between acquisitions is rare. Combined with IReSISD's ability to register 3+ point clouds, our method would even be suitable for mosaicking (in future studies).

### 4.4. Effect heterogeneous resolutions

To reproduce the mismatched sampling densities common in stereo or laser altimetry DTMs, we create versions of a DTM with varying resolutions and levels of detail. This is achieved through downsampling and box avering[7] one of the two point clouds, as listed in Tab. 4 and Tab. 8 of sup. materials. No elevation noise is added. Since we adopt a voxel size at the finest of the two DTM resolutions, the registration result is relative to this resolution.

TEASER++ relies on pairwise distance invariants that are violated when DTMs with same footprints have different resolutions, leading to systematic crash. CPD keeps a voxel-level accuracy despite heavy downsampling. This may seem to contradict previous results were CPD struggled with missing data. However, heterogeneous resolutions preserve the footprints. Besides, the spatially uniform point distribution continues to support CPD's GMM fitting.

IReSISD-DTM also keeps a voxel-level performance on medium downsampling. It sees an increase in error (up to 7.3±5.7 voxels) for the strongest factor 10, while still remaining more accurate than baselines (see sup. materials).

### 4.5. Validation on real data

We first experiment with CTX and MOLA data. The large difference in spatial resolution, by a factor ∼ 17, combined with the noise present in real observations, makes this application particularly challenging. We approximate a registration ground-truth by selecting a terrain for which the two DTMs are visually in good agreement (Fig. 3), and we introduce a reasonable artificial misalignment of norm 5 voxels, using 10 different instances of random translations.

CPD and IReSISD-DTM achieve comparable errors of $2.7 \pm 0.7$ voxels with $0.0 \pm 0.0°$, and $2.1 \pm 0.8$ voxels with $0.9 \pm 0.4°$, respectively. These results confirm the difficulty of large resolution discrepancies. When visually assessing the registration results, both the crater and canyon structures are properly aligned in most instances. Larger errors tend to occur when the random misalignment is oriented along major terrain features, such as canyons, which increases the likelihood of convergence to local minima.

We also visually assess the registration of CaSSIS and CTX data with noticeable (unknown) initial misalignment

---

[7]We use the implementation from Open3D https://github.com/isl-org/Open3D

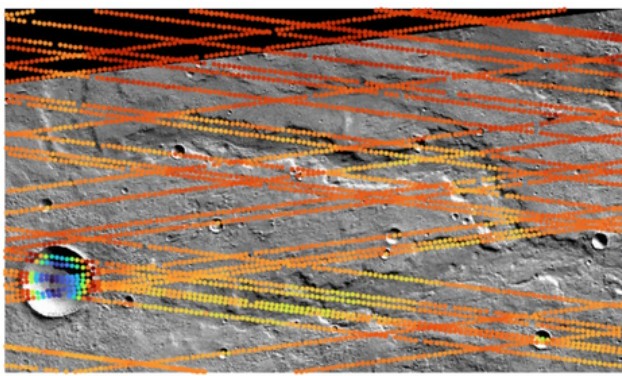

Figure 3. CTX DTM (gray, shaded rendering) and MOLA points (color denotes altitude) experiment data with what we consider groundtruth registration.

and very different footprints, as shown in Fig. 4. The difference in spatial resolution is a small factor $\sim 2$. ICP does not significantly improve DTMs' alignment (Fig. 4b). The partial overlap puts CPD in difficulty as it does its best to align footprints (Fig. 4c). Our method achieves the best visual correspondence between the DTMs, with better alignment of the topagraphic contour lines (Fig. 4d). Eight manually identified landmarks yield average 3D distances of 8.65, 9.20, 43.20, and 7.00 voxels for the original misalignment, ICP, CPD, and IReSISD-DTM respectively, with voxel precision limited by the resolution gap between the two DTMs.

These results confirm the conclusions from experiments on synthetic data. ICP struggles with low availability of salient 3D features, making it unsuitable for general planetary applications. The strong reliance of CPD on footprint matching is also a strong limitation. On the other hand, our method's robustness to different footprints and to different resolutions, due to exploiting full 3D information, makes it suitable for real applications.

## 5. Discussion and Perspectives

We presented IReSISD-DTM, a 3D rigid registration method adapted to planetary remote sensing. To address the lack of salient points, it tackles the registration problem from the point of view of 3D data known as DTMs, rather than 2D images. It leverages and adapts the implicit interpolation of IReSISD to help with challenges associated to missing data and heterogeneous resolutions, while exploiting full 3D geometric information in overlapping regions.

Our method was successfully applied to synthetic and real Martian remote sensing data. It was comparatively evaluated on a new benchmark dataset of synthetic DTMs, against baseline and SOTA 3D registration methods. In the easiest cases with identical DTM footprints, it proved marginally less accurate than the best competitor, although

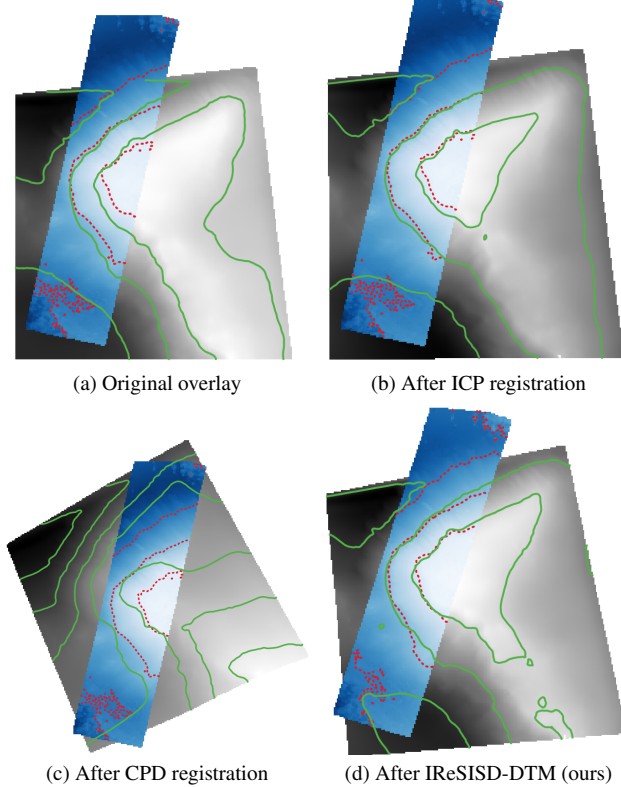

|  |  |
|---|---|
| (a) Original overlay | (b) After ICP registration |
| (c) After CPD registration | (d) After IReSISD-DTM (ours) |

Figure 4. Comparison of registration results on real CaSSIS (blue, 12 m/pixel) and CTX (gray, 6 m/pixel) DTMs with partial overlap. Color intensity encodes elevation. Dotted red (CaSSIS) and green (CTX) contour lines were generated every 10 pixels with GDAL[8] contour (QGIS).

with a perfectly decent voxel-level and often sub-angular accuracy. In the more plausible and challenging scenarios, IReSISD-DTM was significantly more robust to different footprint shapes. This makes it more suitable for real-life operations, especially for the use case of DTM mosaic building where overlap is particularly low. Experiments on real data with moderate noise, heterogeneous resolutions, and different footprints, confirmed these conclusions.

While registration-only methods vary widely in computational cost (e.g. one order of magnitude between ICP and CPD), our method incurs an inherently greater overhead (one further order of magnitude), as it uniquely produces a fused geometric model as a by-product.

Future work should explore the complex registration scenarios of simultaneous alignment of 3+ DTMs in the presence of data gaps and partial overlaps. This requires special attention to memory and time management. A relevant example application is the Oxia Planum region on Mars, where datasets acquired by multiple instruments exhibit these challenges. Applications to other planetary bodies and to Earth remote sensing would also be of interest.

# 6. Acknowledgments

This research was funded by Région Sud through the "Emploi Jeunes Doctorants" program and by the National Program for Space Remote Sensing (PNTS).

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

# 3D registration of remote sensing data for planetary exploration

## Supplementary Material

## 7. Specifications on the hyperparameters of the experiments

We use the standard and advised hyperparameters for every algorithm. In the experiments with the synthetic dataset, we test the methods with large misalignments in order to assess their limits of performance. For our volume's resolution, we adopt the resolution of the finest DTM. Thus, we set $\gamma = 1$ because this volume resolution implies that $C$ requires little or no interpolation. IReSISD-DTM is run for 30 iterations by default, which is a broad boundary that ensures convergence.

When experimenting with the synthetic DTMs, we generate 30 stochastic realizations of terrains for each tested parameter value. We set realistic terrain parameters with an elevation of $\sigma_{ele} = 10$ voxels and elevation noise at a low value ($\sigma_{\mathrm{noise}} = 1$ voxel). For each realization, we duplicate the terrain and apply an optional perturbation (see Sec. 3.1) and a random misregistration to the copy. Misregistration is implemented by a translation vector of fixed norm of 20 voxels, distributed randomly across the $x$, $y$, and $z$ components. This large amount of misalignment for remote sensing data aims at establishing the limits of performance of compared methods. Rotation misregistration is set to zero, as the main errors in real data are expected to be mostly translational. However, registration is performed for both translation and rotation, without any restriction on the rotational DoF.

## 8. Supplementary tables

| $\sigma_{\mathrm{ele}}$ | ICP | RANSAC | CPD | TEASER++ | IReSISD-DTM |
|---|---|---|---|---|---|
| 2 | $16.7 \pm 2.0$ | $16.4 \pm 1.8$ | $\mathbf{1.5 \pm 2.2}$ | $247.0 \pm 150.6$ | $3.2 \pm 2.6$ |
|  | $0.00 \pm 0.00$ | $0.00 \pm 0.00$ | $\mathbf{0.00 \pm 0.00}$ | $0.19 \pm 0.67$ | $0.73 \pm 0.62$ |
| 4 | $16.3 \pm 1.7$ | $16.0 \pm 1.9$ | $1.5 \pm 3.5$ | $267.9 \pm 144.8$ | $\mathbf{1.5 \pm 2.6}$ |
|  | $0.00 \pm 0.00$ | $0.00 \pm 0.00$ | $0.00 \pm 0.00$ | $0.04 \pm 0.60$ | $\mathbf{0.36 \pm 0.63}$ |
| 6 | $16.4 \pm 2.0$ | $16.4 \pm 2.0$ | $\mathbf{0.5 \pm 0.2}$ | $324.4 \pm 96.5$ | $1.1 \pm 0.4$ |
|  | $0.00 \pm 0.00$ | $0.00 \pm 0.00$ | $\mathbf{0.00 \pm 0.00}$ | $0.19 \pm 0.66$ | $0.22 \pm 0.67$ |
| 8 | $16.3 \pm 1.8$ | $16.4 \pm 1.7$ | $1.6 \pm 4.7$ | $279.6 \pm 127.4$ | $\mathbf{1.0 \pm 0.5}$ |
|  | $0.00 \pm 0.00$ | $0.00 \pm 0.00$ | $0.00 \pm 0.00$ | $0.25 \pm 0.66$ | $\mathbf{0.07 \pm 0.68}$ |
| 10 | $16.3 \pm 1.3$ | $16.3 \pm 2.1$ | $\mathbf{0.5 \pm 0.2}$ | $283.0 \pm 125.0$ | $1.0 \pm 0.3$ |
|  | $0.00 \pm 0.00$ | $0.00 \pm 0.00$ | $\mathbf{0.00 \pm 0.00}$ | $0.31 \pm 0.67$ | $0.19 \pm 0.74$ |
| 12 | $17.1 \pm 2.2$ | $16.1 \pm 2.1$ | $\mathbf{0.5 \pm 0.2}$ | $280.0 \pm 109.8$ | $1.7 \pm 3.4$ |
|  | $0.00 \pm 0.00$ | $0.00 \pm 0.00$ | $\mathbf{0.00 \pm 0.00}$ | $0.05 \pm 0.71$ | $0.01 \pm 0.64$ |
| 14 | $16.3 \pm 1.6$ | $16.3 \pm 1.9$ | $\mathbf{0.5 \pm 0.2}$ | $270.9 \pm 121.7$ | $0.9 \pm 0.4$ |
|  | $0.00 \pm 0.00$ | $0.00 \pm 0.00$ | $\mathbf{0.00 \pm 0.00}$ | $0.17 \pm 0.68$ | $0.43 \pm 0.71$ |
| 16 | $16.4 \pm 2.1$ | $16.7 \pm 2.0$ | $\mathbf{0.6 \pm 0.2}$ | $310.4 \pm 110.5$ | $1.0 \pm 0.5$ |
|  | $0.00 \pm 0.00$ | $0.00 \pm 0.00$ | $\mathbf{0.00 \pm 0.00}$ | $0.16 \pm 0.65$ | $0.48 \pm 0.68$ |
| 18 | $16.5 \pm 2.2$ | $16.4 \pm 1.9$ | $1.8 \pm 5.2$ | $290.7 \pm 131.1$ | $\mathbf{1.0 \pm 0.4}$ |
|  | $0.00 \pm 0.00$ | $0.00 \pm 0.00$ | $0.00 \pm 0.00$ | $0.12 \pm 0.66$ | $\mathbf{0.21 \pm 0.67}$ |
| 20 | $16.7 \pm 2.2$ | $16.9 \pm 1.9$ | $\mathbf{0.6 \pm 0.2}$ | $285.4 \pm 122.5$ | $0.9 \pm 0.4$ |
|  | $0.00 \pm 0.00$ | $0.00 \pm 0.00$ | $\mathbf{0.00 \pm 0.00}$ | $0.15 \pm 0.67$ | $0.49 \pm 0.69$ |
| 22 | $16.5 \pm 2.0$ | $16.5 \pm 1.8$ | $\mathbf{0.6 \pm 0.2}$ | $325.6 \pm 113.9$ | $1.0 \pm 0.6$ |
|  | $0.00 \pm 0.00$ | $0.00 \pm 0.00$ | $\mathbf{0.00 \pm 0.00}$ | $0.08 \pm 0.67$ | $0.46 \pm 0.65$ |
| 24 | $16.5 \pm 2.0$ | $16.6 \pm 2.0$ | $\mathbf{0.6 \pm 0.2}$ | $332.2 \pm 125.3$ | $1.1 \pm 0.5$ |
|  | $0.00 \pm 0.00$ | $0.00 \pm 0.00$ | $\mathbf{0.00 \pm 0.00}$ | $0.12 \pm 0.66$ | $0.61 \pm 0.67$ |
| 26 | $16.2 \pm 1.8$ | $16.0 \pm 2.0$ | $\mathbf{0.7 \pm 0.2}$ | $346.9 \pm 103.8$ | $0.9 \pm 0.3$ |
|  | $0.00 \pm 0.00$ | $0.00 \pm 0.00$ | $\mathbf{0.00 \pm 0.00}$ | $0.14 \pm 0.67$ | $0.13 \pm 0.60$ |
| 28 | $16.4 \pm 2.0$ | $16.2 \pm 2.0$ | $\mathbf{0.9 \pm 1.5}$ | $324.9 \pm 127.9$ | $1.4 \pm 2.6$ |
|  | $0.00 \pm 0.00$ | $0.00 \pm 0.00$ | $\mathbf{0.00 \pm 0.00}$ | $0.07 \pm 0.66$ | $0.30 \pm 0.64$ |
| 30 | $16.2 \pm 2.0$ | $16.6 \pm 1.9$ | $1.5 \pm 3.2$ | $324.8 \pm 137.2$ | $\mathbf{1.0 \pm 0.4}$ |
|  | $0.00 \pm 0.00$ | $0.00 \pm 0.00$ | $0.00 \pm 0.00$ | $0.05 \pm 0.66$ | $\mathbf{0.14 \pm 0.56}$ |

Table 5. Effect of elevation variance $\sigma_{\mathrm{ele}}$ on registration and surface correction. Row 1: translation error (mean $\pm$ std, voxels). Row 2: rotation angle $\theta$ (mean $\pm$ std, degrees).

| Configuration | ICP | RANSAC | CPD | TEASER++ | IReSISD-DTM |
|---|---|---|---|---|---|
| #2   r5   s157 | $17.4 \pm 2.1$ | $16.4 \pm 2.0$ | $1.9 \pm 3.2$ | $293.5 \pm 132.8$ | $\mathbf{1.4 \pm 1.5}$ |
|  | $0.00 \pm 0.00$ | $0.00 \pm 0.00$ | $0.00 \pm 0.00$ | $0.06 \pm 0.67$ | $\mathbf{0.19 \pm 0.64}$ |
| #5   r10   s1571 | $16.1 \pm 1.9$ | $16.2 \pm 1.9$ | $\mathbf{2.5 \pm 4.4}$ | $285.9 \pm 134.2$ | $2.7 \pm 4.3$ |
|  | $0.00 \pm 0.00$ | $0.00 \pm 0.00$ | $\mathbf{0.00 \pm 0.00}$ | $0.27 \pm 0.66$ | $0.12 \pm 0.65$ |
| #20   r5   s1571 | $16.5 \pm 1.9$ | $17.0 \pm 2.1$ | $2.3 \pm 3.3$ | $284.8 \pm 131.3$ | $\mathbf{2.0 \pm 2.7}$ |
|  | $0.00 \pm 0.00$ | $0.00 \pm 0.00$ | $0.00 \pm 0.00$ | $0.26 \pm 0.67$ | $\mathbf{0.08 \pm 0.63}$ |
| #2   r30   s5655 | $16.9 \pm 1.8$ | $16.8 \pm 2.1$ | $5.1 \pm 2.1$ | $307.9 \pm 127.5$ | $\mathbf{3.6 \pm 6.9}$ |
|  | $0.00 \pm 0.00$ | $0.00 \pm 0.00$ | $0.00 \pm 0.01$ | $0.19 \pm 0.67$ | $\mathbf{0.10 \pm 0.64}$ |
| #40   r5   s3142 | $16.3 \pm 2.0$ | $16.2 \pm 1.7$ | $1.7 \pm 0.7$ | $322.1 \pm 118.3$ | $\mathbf{1.5 \pm 1.1}$ |
|  | $0.00 \pm 0.00$ | $0.00 \pm 0.00$ | $0.00 \pm 0.00$ | $0.16 \pm 0.66$ | $\mathbf{0.11 \pm 0.65}$ |
| #10   r20   s12566 | $16.6 \pm 1.8$ | $16.3 \pm 1.9$ | $5.7 \pm 3.3$ | $314.9 \pm 111.6$ | $\mathbf{2.4 \pm 3.4}$ |
|  | $0.00 \pm 0.00$ | $0.00 \pm 0.00$ | $0.00 \pm 0.01$ | $0.15 \pm 0.66$ | $\mathbf{0.13 \pm 0.63}$ |
| #1   r80   s20106 | $16.0 \pm 1.9$ | $16.7 \pm 2.0$ | $54.3 \pm 35.4$ | $298.6 \pm 125.6$ | $\mathbf{3.9 \pm 5.4}$ |
|  | $0.00 \pm 0.00$ | $0.00 \pm 0.00$ | $0.04 \pm 0.13$ | $0.10 \pm 0.65$ | $\mathbf{0.05 \pm 0.63}$ |
| #1   r120   s45239 | $16.2 \pm 1.7$ | $16.3 \pm 2.0$ | $162.7 \pm 101.2$ | $298.1 \pm 145.1$ | $\mathbf{4.4 \pm 6.1}$ |
|  | $0.00 \pm 0.00$ | $0.00 \pm 0.00$ | $0.18 \pm 0.44$ | $0.03 \pm 0.60$ | $\mathbf{0.17 \pm 0.61}$ |

Table 6. Effect of missing data on registration and surface correction. Row 1: translation (mean $\pm$ std, voxels). Row 2: rotation angle $\theta$ (mean $\pm$ std, degrees).

| Overlap % | ICP | RANSAC | CPD | TEASER++ | IReSISD-DTM |
|---|---|---|---|---|---|
| 100 | $16.63 \pm 1.75$ | $16.54 \pm 1.97$ | $2.56 \pm 5.79$ | $315.88 \pm 106.04$ | $\mathbf{0.97 \pm 0.38}$ |
|  | $0.00 \pm 0.00$ | $0.00 \pm 0.00$ | $0.00 \pm 0.00$ | $0.29 \pm 0.66$ | $\mathbf{0.61 \pm 0.70}$ |
| 80 | $15.97 \pm 1.79$ | $16.48 \pm 2.14$ | $15.69 \pm 3.14$ | $297.01 \pm 115.00$ | $\mathbf{1.01 \pm 0.38}$ |
|  | $0.00 \pm 0.00$ | $0.00 \pm 0.00$ | $0.00 \pm 0.00$ | $0.25 \pm 0.68$ | $\mathbf{0.37 \pm 0.65}$ |
| 60 | $16.59 \pm 1.88$ | $16.50 \pm 1.65$ | $46.97 \pm 5.45$ | $248.71 \pm 121.48$ | $\mathbf{1.04 \pm 0.45}$ |
|  | $0.00 \pm 0.00$ | $0.00 \pm 0.00$ | $0.00 \pm 0.01$ | $0.62 \pm 0.65$ | $\mathbf{0.26 \pm 0.68}$ |
| 40 | $16.20 \pm 1.94$ | $16.75 \pm 2.02$ | $78.90 \pm 5.87$ | $282.71 \pm 84.71$ | $\mathbf{1.62 \pm 3.20}$ |
|  | $0.00 \pm 0.00$ | $0.00 \pm 0.00$ | $0.00 \pm 0.02$ | $0.03 \pm 0.64$ | $\mathbf{0.34 \pm 0.67}$ |
| 20 | $17.08 \pm 1.96$ | $16.34 \pm 2.00$ | $106.25 \pm 4.34$ | $269.14 \pm 97.47$ | $\mathbf{1.63 \pm 2.57}$ |
|  | $0.00 \pm 0.00$ | $0.00 \pm 0.00$ | $0.01 \pm 0.02$ | $0.09 \pm 0.66$ | $\mathbf{0.39 \pm 0.65}$ |

Table 7. Effect of overlap on registration and surface correction. Rows per overlap: (1) translation error (mean $\pm$ std, voxels), (2) rotation magnitude $\theta$ (mean $\pm$ std, degrees),

| Downsampling factor | ICP | RANSAC | CPD | IReSISD-DTM |
|---|---|---|---|---|
| 2 | $15.829 \pm 1.983$ | $16.423 \pm 1.958$ | $\mathbf{0.495 \pm 0.165}$ | $0.864 \pm 0.220$ |
|  | $0.0017 \pm 0.0031$ | $0.0000 \pm 0.0000$ | $\mathbf{0.0004 \pm 0.0007}$ | $0.2196 \pm 0.6588$ |
| 5 | $15.414 \pm 2.254$ | $16.192 \pm 2.029$ | $\mathbf{0.566 \pm 0.258}$ | $1.627 \pm 0.162$ |
|  | $0.0004 \pm 0.0012$ | $0.0000 \pm 0.0000$ | $\mathbf{0.0003 \pm 0.0003}$ | $0.6294 \pm 0.7346$ |
| 10 | $16.032 \pm 1.863$ | $17.056 \pm 1.744$ | $\mathbf{1.494 \pm 0.439}$ | $7.281 \pm 5.738$ |
|  | $0.0006 \pm 0.0019$ | $0.0000 \pm 0.0000$ | $\mathbf{0.0003 \pm 0.0008}$ | $0.7545 \pm 0.6887$ |

Table 8. Effect of resolution on registration and surface reconstruction. Row 1: translation error (mean $\pm$ std, voxels). Row 2: rotation angle $\theta$ (mean $\pm$ std, degrees).