# OpenReview forum: "3D registration of remote sensing data for planetary exploration"
_thecvf.com/CVPR/2026/Workshop/3D4S — CVPR 2026 Workshop 3D4S Poster_

### Official Review · Reviewer_pLbH · 2026-04-25
**This paper proposes IReSISD-DTM, a 3D geometric method for rigid registration of Digital Terrain Models (DTMs) in planetary remote sensing. The approach jointly performs registration and geometric modeling using a level-set formulation, where multiple DTMs are aligned to a shared implicit surface instead of relying on explicit point correspondences. The method is evaluated on synthetic benchmarks and real Martian datasets, showing improved robustness in challenging scenarios such as missing data, low overlap, and heterogeneous resolutions.**

**Rating:** 6
**Confidence:** 3

**Review:**

### **Pros**

- **Well-motivated problem with clear practical relevance.**
  The paper addresses real challenges in planetary data (e.g., lack of salient features, partial overlap), which are clearly explained and important for large-scale remote sensing applications.

- **Strong robustness in challenging scenarios.**
  Experiments (e.g., Tables 2 and 3) show the method performs consistently under missing data and low overlap, where baselines like CPD degrade significantly (e.g., large errors under low overlap).
---

### **Cons**

- **Limited novelty beyond adapting existing framework.**
  The method is largely an extension of prior IReSISD work, with modifications tailored to DTMs. The core formulation (level-set + joint registration/modeling) is not fundamentally new.

- **Evaluation somewhat biased and incomplete.**
  The paper explicitly excludes learning-based methods and lacks comparison with modern deep registration approaches. In addition, real-data evaluation relies mostly on qualitative or approximate ground truth, making it hard to assess actual performance.

- **Claims of superiority are not fully supported.**
  While the method is more robust in some cases (e.g., large gaps), it is often **less accurate than CPD in easier settings** (Table 1), which weakens the overall claim of being SOTA.

---

### Official Review · Reviewer_X2cE · 2026-04-25
**3D Registration of Remote Sensing Data for Planetary Exploration**

**Rating:** 7
**Confidence:** 3

**Review:**

This paper presents a 3D rigid registration method, IReSISD-DTM, for aligning planetary remote sensing Digital Terrain Models (DTMs). The work targets an important and practical problem in planetary exploration, where remote sensing acquisitions often suffer from limited overlap, different footprints, missing data, heterogeneous resolutions, and a lack of salient visual landmarks. The paper is well motivated, and the proposed adaptation of IReSISD to open-boundary DTM surfaces is technically reasonable. The introduction of a synthetic planetary DTM benchmark and the validation on real Martian data further strengthen the paper’s relevance.

The main strength of the paper is its clear focus on DTM-specific challenges rather than treating planetary terrain registration as a generic point-cloud registration problem. The proposed method shows strong robustness under missing-data and partial-overlap conditions, where standard baselines such as ICP, RANSAC, CPD, and TEASER++ struggle. The experimental results are generally convincing for the target application, especially the low-overlap and different-footprint cases.

However, there are several aspects that could be improved. First, the experimental setup mainly introduces translational misalignment, while the method is described as full rigid registration; additional experiments with controlled rotational perturbations would better support the 6-DoF claim. Second, the real-data validation is useful but mostly qualitative, since true ground-truth registration is unavailable. More objective validation criteria, such as landmark-based or contour-based alignment measures, would strengthen the conclusions. Third, the paper emphasizes scalability and potential use for multi-DTM mosaicking, but the experiments remain mostly pairwise and do not include runtime or memory analysis. Finally, the exclusion of learning-based methods is understandable for this domain, but the discussion could be more nuanced.

Overall, this is a solid workshop paper with a clear practical motivation, a reasonable technical contribution, and promising experimental results. While the validation is not exhaustive, the method appears well suited to the specific challenges of planetary DTM registration and could be valuable for future remote sensing and planetary exploration workflows.

---

### Decision · Program_Chairs · 2026-04-28

Accept (Poster)